# The Use of Executive Fluency Tasks to Detect Cognitive Impairment in Individuals with Subjective Cognitive Decline

**DOI:** 10.3390/bs12120491

**Published:** 2022-12-02

**Authors:** Joël Macoir, Pascale Tremblay, Carol Hudon

**Affiliations:** 1Faculté de Médecine, Département de Réadaptation, Université Laval, Québec, QC G1V 0A6, Canada; 2Centre de Recherche CERVO—Brain Research Centre, Québec, QC G1J 2G3, Canada; 3Faculté des Sciences Sociales, École de Psychologie, Université Laval, Québec, QC G1V 0A6, Canada; 4Centre de Recherche VITAM, Québec, QC G1J 2G1, Canada

**Keywords:** subjective cognitive decline, Alzheimer’s disease, verbal fluency, executive functions, assessment, cognition

## Abstract

Objective: Although evidence has indicated that subjective cognitive decline (SCD) may be an early sign of Alzheimer’s disease (AD), the objectification of cognitive impairment in SCD is challenging, mainly due to the lack of sensitivity in assessment tools. The present study investigated the potential contribution of two verbal fluency tasks with high executive processing loads to the identification of cognitive impairment in SCD. Methods: A total of 60 adults with SCD and 60 healthy controls (HCs) performed one free action (verb) fluency task and two fluency tasks with more executive processing load—an alternating fluency task and an orthographic constraint fluency task—and the results were compared. Result: In the free action fluency task, the performance of the participants with SCD and the HCs was similar. However, HCs performed significantly better than SCD in the alternating fluency task, which required mental flexibility, and the orthographic constraint fluency task, which required inhibition. Discussion: The study findings suggest that verbal fluency tasks with high executive processing load could be useful in detecting cognitive deficits at the preclinical stage of AD. The inclusion of such tests in assessment batteries should be considered in order to improve the detection of subtle cognitive impairment in preclinical major neurocognitive disorder populations.

## 1. Introduction

Low birth rates and advancing life expectancy are contributing to population aging in industrialized countries, such as the United States and Canada. For example, Statistics Canada reported a significant increase in recent years in the number of people aged 65 years and over [1]. Furthermore, this group accounted for about 18.5% of Canada’s population in 2021 [1]. The incidence of age-related diseases, including major neurocognitive disorders (MNCD), has also increased dramatically [2]. According to the DSM-V, MNCD refers to a progressive decline in cognitive function over time that affects daily life activities to a significant degree [3]. Alzheimer’s disease (AD) is the main etiology of MNCD.

The typical course of MNCD due to AD includes three main phases: (1) a pre-clinical phase in which the individual can be placed on a continuum from completely asymptomatic to a very subtle decline known as subjective cognitive decline (SCD) or subjective cognitive impairment; (2) mild cognitive impairment (MCI), which is a pre-MNCD phase characterized by memory impairment or other deficits of cognition, and (3) MNCD [4,5]. Typically diagnosed among the elderly, MNCD is characterized by a significant decline in cognitive function that impacts daily activities and social interactions. The cognitive impairment associated with MNCD affects the following domains: learning and memory, language, executive function, complex attention, perceptual-motor function, and social cognition.

The pre-symptomatic phase of AD and other major forms of dementia can last for several decades [6]. Moreover, the self-reporting of subtle cognitive problems has been associated with an increased likelihood of dementia [7,8]. According to Jessen et al., SCD has two major features: (1) a self-experienced decline in cognitive function, compared to a previous normal state, that is unrelated to an acute event and not necessarily corroborated by relatives, and (2) unimpaired performance on standardized cognitive tests [9]. Jessen et al. also introduced the concept of SCD *plus*, which is characterized by the following clinical features that increase the risk of future decline to MCI or dementia: SCD in memory irrespective of the individual’s functional abilities in other cognitive domains; onset of SCD within the past 5 years; onset of SCD in individuals aged 60 years and older; expression of concerns associated with SCD; persistence of SCD over time; the seeking of medical help because of SCD, and; the confirmation of cognitive decline by an observer (i.e., close relatives, caregivers) (9).

Most people aged over 65 experience at least occasional cognitive complaints [10]. In population-based studies, the prevalence of SCD varied between 10% and 60% in individuals aged 70 years and older [11]. Although the progression from SCD to MNCD is not systematic, longitudinal epidemiological studies have revealed an association between SCD and a significantly increased risk of progression to MCI and dementia over the years [12,13]. Moreover, studies have identified an association between SCD and depressive symptoms and suggested that the latter is a confounding factor in the relationship between SCD and progression to MNCD [14,15].

The early detection of neurodegenerative diseases is a critical concern for public health and clinical research focused on dementia prevention. The diagnostic of AD and MCI is based on consensual criteria, including recommendations for the assessment of cognitive function [16,17]. In SCD however, the lack of sensitivity in assessment tools makes it challenging to detect cognitive deficits, as individuals can compensate for performance deficits and normalize their performance [4]. A systematic neuropsychological assessment of individuals with SCD is certainly not recommended in clinical practice; however, early identification of the detrimental effects of AD pathology on cognitive abilities is crucial for counseling interventions, prevention, and care optimization.

Most studies on the objectification of cognitive differences between individuals with SCD and healthy controls (HCs) have focused on the predictive value of cognition questionnaires. For example, Maruta and Martins showed that individuals with language complaints performed significantly worse on semantic fluency tasks than the HCs after five years, even though their rate of decline was similar overall and unrelated to subsequent cognitive decline [18]. Based on the findings of a six-year longitudinal study, Brailean et al. argued that a decrease in subjective memory performance that cannot be explained by depressive symptoms may be detected by self-reported memory complaints [19]. Meanwhile, a recent study of a small group of individuals with SCD revealed an association between performance on demanding cognitive tasks and self-perceived memory functioning [20].

A few cross-sectional studies have aimed to identify objective impairment in SCD using tasks with a high cognitive processing load. With respect to memory, Park et al. identified significant interactions between subjective memory complaints and performance on verbal episodic memory tasks in a sample of 219 Korean people aged 55 years and older [21]. A similar relationship between SCD and delayed recall in verbal episodic memory was reported in a sample of individuals with a few depressive symptoms [22]. Furthermore, a decline in performance in tests exploring associative memory, memory binding, prospective memory, and visual short-term memory has been found in individuals with SCD [23,24,25,26,27]. 

The objectification of the poorer performance of individuals with SCD compared to that of the HCs has not been limited to memory function. Promising results have emerged from studies using verbal fluency tests, which explore lexical access and executive functions. For example, a few studies have shown that, compared to the HCs, individuals with SCD generated significantly fewer words during semantic and/or orthographic verbal fluency tasks [28,29,30]. In addition, we recently identified a continuum of performance on a free action (verb) fluency task; the HCs performed better than the participants with SCD, who, in turn, performed better than the participants with MCI [31]. In other studies, however, the authors did not find any detectable difference in the performance of individuals with SCD and healthy controls in visuoconstructional tests [32] or in picture naming and face identification [33].

Minor executive function deficits have also been reported in patients with SCD [30,34]. Impaired executive function is considered an early sign of cognitive decline in the course of pathological aging [34]. In a recent systematic review, Webster-Cordero and Giménez-Llort found a relationship between cognitive complaints and early impairment of executive functions (i.e., working memory, initiative, cognitive flexibility, inhibition, planning) associated with AD cerebrospinal fluid biomarkers and reduced cortical volume in bilateral hippocampi and left frontal regions [35].

Compensatory mechanisms in individuals with SCD [36,37] make it difficult to detect cognitive decline using conventional assessment tools [38]. In light of this, the use of neuropsychological tasks with high processing loads seems promising because of their potential to lessen compensation mechanisms or render those mechanisms ineffective. The main objective of the present study was to determine the potential contribution of verbal fluency tasks with high executive processing loads (i.e., alternating fluency and orthographic constraint fluency) to the identification of cognitive impairment in SCD.

## 2. Methods

### 2.1. Participants

The sample population of the present study consisted of 60 adults with SCD and 60 HCs. The participants, who ranged in age from 55 to 75 years old and spoke French as their primary language, were recruited through advertisements in the community. All the participants with SCD were worried about their cognitive functioning, and they met Jessen et al.’s aforementioned criteria for SCD [4]. In the present study, all participants with SCD, except for one 56-year-old individual, were 60 years old and older. Unfortunately, the number of participants with SCD who fulfilled the SCD *plus* criteria could not be ascertained, since this study began before their publication [9].

A self-reported health questionnaire was used to confirm the good physical and mental health of all HCs. They did not report any SCD-like subjective cognitive complaints, and they performed above the cutoff on the Montreal Cognitive Assessment (MoCA) [39], which was established for the Quebec-French population taking into account age, education, and sex [40]; these results suggested the preservation of general cognitive functioning.

Individuals with a history of moderate or severe traumatic brain injury, cerebrovascular disease, delirium (in the past six months), intracranial surgery, a neurological disorder of cerebral origin, and/or encephalitis or bacterial meningitis, as well as those who had oncological treatments (in the past 12 months) or general anesthesia (in the past six months), were excluded from the study. Additional exclusion criteria included an unstable metabolic or medical condition (e.g., untreated hypothyroidism or diabetes), a history or actual diagnosis of a psychiatric disorder according to the DSM-V (Axis I) [3], alcoholism or substance abuse (in the past 12 months), uncorrected vision or hearing problems, the use of experimental medication, and the inability to provide informed consent. The information regarding exclusion criteria was obtained from participants’ self-reports.

Written informed consent was obtained from all participants at the beginning of the study in accordance with the Declaration of Helsinki. The study was approved by the local research ethics board (Ethics committee on sectoral research in neurosciences and mental health of the CIUSSS de la Capitale-Nationale (project number 2019-1529).

### 2.2. Clinical Assessment and Group Characterization

All participants completed a comprehensive set of clinical tests in order to validate the inclusion/exclusion criteria and classify them into the SCD and HC groups. The battery of tests included measures of cognitive complaints, depressive symptoms, anxiety symptoms, general cognitive status, and general language status.

The Questionnaire de Dépistage de la Plainte Cognitive (Screening Questionnaire of Cognitive Complaints; QDPC) Dion et al., Unpublished) is a simple and easy-to-administer questionnaire for cognitive complaints [41]. The QDPC uses the following questions and sub-questions to address an individual’s cognitive decline in comparison to his/her former level of functioning as well as his/her cognitive function as compared to other people of the same age group: Are you worried about how your memory is working?Do you think your memory has changed in the last 10 years?
2.1If yes, how long have you observed a decline in memory functioning?Do you feel that your memory is worse than that of other people your age?
3.1If yes, and it is worse, do you feel that you have always had a poorer memory than other people your age?3.2If no, and it is the same, would you say that, in the past, your memory was at the same level as or better than most other people your age?

Based on the Jessen et al. criteria [4], the participants were categorized as having SCD if they responded yes to Questions 2, 3, and 3.1. They were also placed in this category if they responded yes to Question 2, no to Question 3, and noted that their memory was better than most other people their age in response to Question 3.2.

Since depressive and anxiety symptoms are frequently associated with SCD [14], all participants were evaluated using the 30-item Geriatric Depression Scale (GDS-30) [42] and the Geriatric Anxiety Inventory (GAI) [43]. These detection scales, which are based on yes-no questions, can be used to identify depressive and anxiety symptoms, respectively, in the elderly. General cognitive impairment was identified using the MoCA [39,40]. This widely used screening test, which was designed to detect cognitive impairment associated with MCI, was shown to be sensitive to mild cognitive deficits and able to predict future cognitive decline in several cognitively impaired states, including AD and other forms of MNCD [44]. The verbal fluency tasks used in the present study not only involved executive functions but also various language processes. Possible language deficits were identified using the Detection Test for Language impairments in Adults and the Aged (DTLA) [45], a quick screening test that specifically addresses the language impairment encountered in most neurodegenerative diseases.

The demographic data and clinical tests results are shown in Table 1. Analyses of variance (ANOVAs) were used to compare the groups based on demographic data, except for sex, which was analyzed using the chi-square test. The groups were equivalent with respect to age and sex; however, the participants with SCD were more educated than the HCs. All participants performed within the normal range on the MoCA based on normative data that considered age, education, and sex [40], and no difference was found between the groups. DTLA performance was equivalent in the groups, suggesting normal language functioning. The GDS-30 scores of the HCs and the participants with SCD ranged from 0 to 16 and 0 to 20, respectively. None of the participants had clinical depression, according to the DSM-V criteria [3]. However, there was a significant difference in the GDS-30 scores of the two groups. The GAI scores of the HCs and the participants with SCD ranged from 0 to 17 and 0 to 14, respectively. None of the participants had clinical anxiety, according to the DSM-V criteria [3]. The groups’ GAI scores were statistically equivalent.

### 2.3. Experimental Tasks

All participants performed free action (verb) fluency, alternating fluency, and orthographic constraint fluency tasks. The purpose of these verbal fluency tasks was to contrast the participants’ performance on tasks with low (free action fluency) and high (alternating and constraint fluency) levels of executive processing load.

Free action fluency. The free action fluency task requires less involvement of executive functions than the two other verbal fluency tasks described below [46]. The participants were given the following instructions in French:

I would like you to tell me as many different things as you can think of that people do. You must produce single words, such as eat or drink, rather than sentences. However, you cannot produce the same verb with different endings, such as eat, ate, and eaten. Can you give me an example of something that people do?

If the response was deemed unacceptable, the participants were asked to provide another example of an action word (any verb response was acceptable). If that response was acceptable, the examiner said, “Good. Now, to avoid distractions, please close your eyes and tell me, in one minute, as many different things as you can think of that people do.”.

The scoring method was based on the number of new verbs produced in one minute, the number of errors (i.e., repeated verbs and verbs not respecting the fluency criteria), and the number of verbs produced within each time interval (1: 1–29 s and 2: 30–60 s).

Alternating fluency. Alternating fluency tasks require more mental flexibility than traditional fluency tasks [47]. The participants were asked to alternately produce words beginning with the letter T and words belonging to the clothes category. They were given the following instructions in French:

For 90 s, I would like you to name, alternately, words that start with the letter T (excluding proper names) and words that belong to the category of clothing. Therefore, you must tell me a word that starts with T, then name a clothing item, then name a word that starts with T, and so on. Now, to avoid any distractions, please close your eyes and tell me as many words as possible by alternating between words that start with T and the names of clothing items. Are you ready?

The participants were assessed based on the number of words respecting the alternance, the number of alternance errors, and the number of words produced within each time interval (I1: 1–29 s, I2: 30–59 s, and I3: 60–90 s).

Orthographic constraint fluency. Compared to typical fluency tasks, the orthographic constraint fluency task is assumed to require more inhibitory control abilities. In this new and original task, the participants were asked to produce names of animals whose written form did not involve the letter A. They were given the following instructions in French:

For 90 s, I would like you to tell me words that belong to the category of animals but do not include the letter A for 90 s. Therefore, you must tell me animal names in which there is no letter A at all. Now, to avoid any distractions, please close your eyes and tell me as many animal names as possible in which there is no letter A. Are you ready?

The participants were assessed based on the number of words respecting the fluency criterion, the number of constraint errors, and the number of words produced within each time interval (I1: 1–29 s, I2: 30–59 s, and I3: 60–90 s).

The participants were assessed in two sessions of 60 min, during which each of the tests for the characterization of the clinical and cognitive profile and the verbal fluency tests were administered in the same order.

### 2.4. Statistical Analyses

All statistical analyses were conducted using Jamovi [48]. First, correlation analysis was performed between the dependent measures (i.e., total responses, total errors, number of words in each time interval) and education level as well as with GDS score, as the two groups were statistically different on these variables. If the data met normal distribution (Shapiro-Wilke > 0.5) and variance homogeneity requirements, Pearson’s correlation was conducted; otherwise, Spearman’s correlation was conducted. If any correlations existed, education level and/or GDS-30 were considered as covariables in the analyses.

Comparisons between the SCD and HC groups in terms of the total number of responses and the total number of errors were analyzed using separate ANOVAs. Next, to explore possible differences among the groups according to the time interval, the dependent measure (i.e., number of words in each time interval) was analyzed using two (i.e., HC and SCD) by two (i.e., free action fluency: 0–30 and 31–60 s) or three (i.e., alternating and orthographic constraint fluency: 0–30, 31–60, and 61–90 s) mixed model ANOVAs with repeated measures on the second factor. Pairwise comparisons were computed using paired *t*-tests with a Bonferroni correction. The significance level was set at 0.05 for the ANOVA and post hoc tests. The effect sizes were reported as partial eta squared (η2) and interpreted according to Richardson [49] (i.e., 0.01, 0.06, and 0.14 indicated small, medium, and large effects, respectively).

## 3. Results

### 3.1. Free Action Fluency

Table 2 shows the mean, SD, and range results for the free action fluency task by group and time interval. No correlations were found between the dependent measures and education level (all *p* > 0.30) or between those measures and GDS-30 (all *p* > 0.15). Furthermore, there were no significant differences between the SCD and HC groups in terms of the total number of correct responses or the number of errors. The mixed model ANOVA examining the total number of words in each time interval revealed a main effect for the time interval, *F*(1, 118) = 147.09, *p* < 0.001. The post hoc test indicated that the participants produced significantly more words in the first time interval (*M* = 11.5) than in the second one (*M* = 8.3), *p* < 0.001. The group by time interval interaction was not significant, *F*(1, 118) = 6.56, *p* = 0.012. All errors made by the participants consisted of repetition of verbs that had already been produced.

### 3.2. Alternating Fluency

Table 3 shows the mean, SD, and range results for the alternating fluency task by group and time interval. No correlations were found between the dependent measures and education level (all *p* > 0.33) or between those measures and GDS-30 (all *p* > 0.13). There were significant differences in the two groups’ alternating fluency task results. Performance was significantly lower in the SCD group than in the HC group in terms of the total number of words respecting the alternance; however, there was no difference in the number of alternance errors. The mixed model ANOVA examining the total number of words in each time interval revealed a main effect for the time interval, *F*(1, 118) = 255.11, *p* < 0.001. The post hoc test indicated that the participants produced significantly more verbs in the first time interval (*M* = 8.83) than in the second (*M* = 5.50) and third (*M* = 4.80) ones, *p* < 0.001. They also produced significantly more words in the second time interval than in the third one, *p* < 0.001. The group by time interval interaction was not significant, *F*(1, 118) = 1.74, *p* = 0.18.

### 3.3. Orthographic Constraint Fluency

Table 4 shows the mean, SD, and range results for the orthographic constraint fluency task by group and time interval. No correlations were found between the dependent measures and education level (all *p* > 0.27) or between those measures and GDS-30 (all *p* > 0.09). There were significant differences in the two groups’ orthographic constraint fluency task results. The performance was significantly lower in the SCD group than in the HC group in terms of the total number of words respecting the constraint; however, there was no difference in the number of constraint errors. The mixed model ANOVA examining the total number of words in each time interval revealed a main effect for the time interval, *F*(1, 118) = 106.72, *p* < 0.001. The post hoc test indicated that the participants produced significantly more words in the first time interval (*M* = 6.19) than in the second (*M* = 3.62) and third (*M* = 3.07) ones, *p* < 0.001. They also produced significantly more words in the second time interval than in the third one, *p* < 0.029. The group by time interval interaction was not significant, *F*(1, 118) = 0.24, *p* = 0.79.

## 4. Discussion

The main objective of the present study was to determine the potential contribution of verbal fluency tasks with high executive processing loads to the identification of cognitive impairment in SCD. In the free action fluency task, which required limited involvement of executive functions, the performance of the SCD and HC groups was similar. However, performances were significantly lower in the SCD group than in the HC group in the alternating fluency task, which required flexibility, and the orthographic constraint fluency task, which required inhibition. In those tasks, the lower performance level of the participants with SCD was not attributable to differences among the groups in terms of education level or depressive symptoms. These results are in line with studies indicating that tasks with a high cognitive processing load may be helpful for detecting impairment in SCD [20,23,24,25,26,27,50,51].

Normal aging has been shown to negatively affect executive functions, especially among people older than 70 years [52]. Declines in executive functions have been observed in terms of mental flexibility and response inhibition [53,54,55]. Furthermore, dysexecutive functioning has been reported along the continuum of pathological aging, including MCI and AD [34,56]. The impairment or decline of executive functions has also been reported in SCD, and it is considered an early sign of cognitive decline in pathological aging [30,34,35,57].

Verbal fluency tasks are used in clinical practice and research to measure the speed of lexical access as well as executive functions, especially updating, inhibition, and mental flexibility [58,59]. Impaired performance in verbal fluency has been reported in SCD research in which classical semantic and orthographic fluency tasks were used; however, some studies obtained inconsistent or inconclusive results [28,29,30,60,61,62]. For example, Nutter-Upham et al. reported a HC > SCD > MCI continuum of performance in verbal fluency but did not detect a statistical difference between the participants with SCD and the HCs [29]. Fagundo et al. and Östberg et al. found significant differences in verbal fluency between the participants with SCD, MCI, and AD; however, the absence of a control group in each study made it impossible to assess the decline in SCD [60,61]. Nikolai et al. showed that, compared to the HCs, the participants with SCD generated significantly fewer words in semantic tasks but not in orthographic fluency tasks [28].

In the present study, two executive verbal fluency tasks were used to detect cognitive impairment in SCD. Alternating fluency has proven successful to differentiate healthy individuals from participants with various clinical conditions, such as Parkinson’s disease, MCI, frontotemporal dementia, and frontal lobe lesions [29,47,62,63]. Alternating fluency tasks may be administered with two distinct conditions: intra- and extra-dimensional. In tasks with the intra-dimensional condition, participants are asked to alternate between probes of the same domain (e.g., words corresponding to concepts from two semantic categories, such as fruits and clothes, or words beginning with C and with F). The present study used a task with an extra-dimensional condition in which participants had to alternately generate words from the semantic and orthographic domains. This condition imposes high executive processing demands due to the obligation to alternate between two distinct lexical search methods. This additional executive load was for example demonstrated by Downes et al. who showed that individuals with idiopathic Parkinson’s disease performed at the same level than healthy participants in the semantic, orthographic, and alternating intra-dimensional verbal fluency tasks, while a specific impairment emerge in the extra-dimensional verbal fluency condition [47]. Likewise, in the present study, the participants with SCD and the HCs performed similarly in the free action fluency task, while the extra-dimensional alternating fluency task revealed differences between these groups.

To the best of our knowledge, the present study is the first one to use an orthographic constraint verbal fluency task. This task included an extra-dimensional condition, since participants had to generate words according to a semantic criterion (e.g., animals) while inhibiting responses according to a lexical criterion (e.g., animal names written with the letter A). This task is particularly difficult in a language such as French due to its deep orthography—that is, the lack of one-to-one mapping in phoneme-to-grapheme correspondences. For example, in the constraint fluency task, the animal names *corbeau* [kɔʀbo] (crow) and *kangourou* [kᾶguʀu] (kangaroo) must be inhibited since their written forms contain the letter A, even though their spoken forms do not include the corresponding phoneme ([a] or [ɑ]). In these words, the letter is embedded in a multi-letter grapheme (i.e., *eau* [o] in *corbeau*; *an* [ᾶ] in *kangourou*). Studies have shown that the graphemic complexity of words plays a role in word recognition and production [64,65]. Although the performance of the SCD and HC groups in the present study did not differ in terms of the number of constraint errors, one cannot rule out the possibility that graphemic complexity contributed to the differences observed between these groups in terms of the number of words produced, while respecting the orthographic constraint.

According to longitudinal studies, SCD and MCI are associated with a similar increased risk of AD [66,67]. In a meta-analysis of longitudinal studies on the conversion rate of SCD to dementia, Mitchell et al. found that individuals with SCD who had no objective deficits were twice as likely to develop dementia as individuals without SCD [68]. Moreover, studies have found an association between SCD and β-amyloid (Aβ) burden in the brain as well as in cerebrospinal fluid [69,70]. However, there is controversy on this topic, with other studies reporting null findings or showing that brain β-amyloidosis alone did not predict progression to MCI or AD [71,72]. Hypotheses to explain these inconsistent results notably point to heterogeneity in the SCD population regarding cognitive reserve and its association with psychiatric symptoms, such as depression and anxiety [73,74,75]. In the present study, there was no difference between the SCD and HC groups regarding the presence of anxiety symptoms. Moreover, the impaired performance of the participants with SCD in executive fluency tasks could not be attributed to the fact that they had more depressive symptoms than the HCs.

The assessment tools used to assess SCD vary greatly across studies (e.g., interviews, questionnaires, general cognitive tests, neuropsychological tests) and may be a potential source of heterogeneity in their results. In the present study, we used verbal fluency tasks that explicitly tapped executive functions, which are fragile in normal aging and impaired in SCD. The high processing load of these tasks imposed additional demands on executive functions and thus allowed for the objectification of cognitive impairment in the SCD group.

Meanwhile, studies have revealed a higher risk of AD associated with SCD in highly educated people than in people with low levels of education [76,77]. A possible explanation for this paradoxical relationship could be explained by the cognitive reserve hypothesis, which posits that cognitive decline is more rapid in individuals with high cognitive reserve due to the delayed onset of clinical symptoms [78]. In other words, cognitive decline develops quickly when the neuroprotective role of the cognitive reserve no longer operates as a compensation mechanism.

The present study had several limitations. First, its cross-sectional design did not allow tracking of the progression of cognitive decline in SCD and, therefore, it could not estimate the predictive value of executive verbal fluency tasks in terms of the progression of SCD to MCI and dementia. Second, it did not include a biomarker confirmation of preclinical AD. Third, although the participants with SCD performed within the normal range on the MoCA and the DTLA, suggesting normal cognitive and language functioning, a more extensive assessment of their neuropsychological and neurolinguistic abilities would have provided a comprehensive characterization of the cognitive processes underlying their impairment in the executive verbal fluency tasks. The fourth limitation stems from the sampling method, which is a well-known source of inconsistency in results [79]. For example, studies have shown that participants with SCD recruited in a memory clinic are more likely to progress to MCI than those recruited in the general population [80,81]. As pointed out by Rodríguez-Gómez et al., population-based samples are more representative of the population with cognitive complaints [79]. Although less biased than a population recruited in a medical setting, the convenience sampling method used in the present study was vulnerable to selection bias.

In conclusion, SCD symptoms are nonspecific and can be found in various clinical conditions, such as frontotemporal dementia, Parkinson’s disease, depression, and bipolar disorder [82,83,84]. Nevertheless, individuals with SCD have an increased risk of AD, justifying the development of assessment tools that are better adapted to early symptoms of cognitive decline. This is particularly important because individuals with SCD employ compensatory mechanisms that make it difficult to detect cognitive decline with conventional screening stools. The study findings suggest that verbal fluency tasks with additional executive processing loads could be useful in detecting cognitive deficits at the preclinical stage of AD. These tasks are simple to use and easy to incorporate into clinical test batteries. Future studies should identify which cognitive domains are most susceptible to be impaired in SCD; establish more formal links between cognitive complaints and cognitive impairments; and develop more sensitive neuropsychological tests of episodic memory, executive functions and language.

## Figures and Tables

**Table 1 behavsci-12-00491-t001:** The demographic and cognitive characteristics of groups.

	HC (n = 60)		SCD (n = 60)				
	*M* (*SD*)	min–max	*M* (*SD*)	min–max	*F*	*p*	*Effect Size*
Age	66.5 (4.99)	55–75	66.6 (4.92)	56–75	3.39 × 10^−4^	0.985	*n*^2^ = 0.000
Education	15.9 (2.58)	11–22	17.7 (3.30)	11–25	11.30	0.001 ***	*n*^2^ = 0.087
Males/females	29/31		28/32		0.033 ^t^	0.85	
MoCA (30)	27.8 (1.59)	24–30	27.1 (1.78)	24–30	3.78	0.54	*n*^2^ = 0.031
DTLA (100)	95.3 (5.79)	77–100	95.6 (5.12)	83–100	0.10	0.75	*n*^2^ = 0.001
GDS (30)	5.55 (4.68)	0–16	8.17 (4.56)	0–20	9.62	0.002 **	*n*^2^ = 0.075
GAI (20)	3.38 (3.94)	0–17	4.35 (4.25)	0–14	1.67	0.20	*n*^2^ = 0.014

Note: *M* = Mean; SD = Standard deviation; min-max = minimal-maximal test score value. DTLA = Detection test for language impairments in adults and the aged; GAI = Geriatric Anxiety Inventory; GDS = Geriatric Depression Scale; HC = Healthy controls; MoCA = Montreal cognitive assessment; SCD = Subjective cognitive decline. *** *p <* 0.001; ** *p* < 0.01; ^t^ = Pearson’s Chi-squared test.

**Table 2 behavsci-12-00491-t002:** The results on the free action fluency task according to Group, Performance and Time Interval.

	HCs (n = 60)			SCD (n = 60)					
Performance	*Mean*	*SD*	*Range*	*Mean*	*SD*	*Range*	*F*	*p*	*Effect Size*
Total response	20.6	5.21	10–33	19.6	6.23	7–37	1.10	0.30	*n*^2^ = 0.009
Total errors	0.62	1.01	0–4	0.45	0.675	0–2	1.13	0.29	*n*^2^ = 0.009
Time interval	*Mean*	*SD*	*Range*	*Mean*	*SD*	*Range*	*t*	*p*	
Interval 1 (1–29 s)	12.3	3.27	6–19	11.1	3.62	3–20	−1.96	0.315	
Interval 2 (30–59 s)	8.33	2.81	2–14	8.48	3.19	3–17	0.27	1	

Note: HCs: healthy controls; SCD: subjective cognitive decline; SD: standard deviation.

**Table 3 behavsci-12-00491-t003:** The results on the Alternating fluency task according to Group, Performance and Time Interval.

	HCs (n = 60)			SCD (n = 60)					
Performance	*Mean*	*SD*	*Range*	*Mean*	*SD*	*Range*	*F*	*p*	*Effect size*
Total response	20.4	3.53	12–28	17.9	4.07	10–29	12.4	<0.001 ***	*n^2^* = 0.095
Alternance errors	0.57	1.23	0–6	0.68	0.85	0–3	0.37	=0.55	*n^2^* = 0.003
Time interval	*Mean*	*SD*	*Range*	*Mean*	*SD*	*Range*	*t*	*p*	
Interval 1 (1–29 s)	9.4	2.19	5–14	8.27	2.00	4–14	−2.96	0.055	
Interval 2 (30–59 s)	5.95	1.60	3–10	5.05	1.69	0–10	−2.995	0.050	
Interval 3 (60–90 s)	5.02	1.32	2–8	4.58	1.59	0–8	−1.63	1	

Note: HCs: healthy controls; SCD: subjective cognitive decline; SD: standard deviation. *** *p* < 0.001.

**Table 4 behavsci-12-00491-t004:** The results on the Constraint fluency task according to Group, Performance and Time Interval.

	HCs (n = 60)			SCD (n = 60)					
Performance	*Mean*	*SD*	*Range*	*Mean*	*SD*	*Range*	*F*	*p*	*Effect size*
Total response	13.6	3.63	4–26	12.1	3.33	7–21	5.69	0.019 *	*n*^2^ = 0.046
Constraint errors	1.03	1.18	0–6	1.20	1.60	0–7	0.42	0.52	*n*^2^ = 0.004
Time interval	*Mean*	*SD*	*Range*	*Mean*	*SD*	*Range*	*t*	*p*	
Interval 1 (1–29 s)	6.37	1.94	2–10	6.02	1.92	2–12	−0.99	1	
Interval 2 (30–59 s)	3.87	1.96	0–9	3.37	1.75	0–7	−1.475	1	
Interval 3 (60–90 s)	3.40	1.88	0–11	2.73	1.66	0–8	−2.06	0.62	

Note: HCs: healthy controls; SCD: subjective cognitive decline; SD: standard deviation. * *p* < 0.05.

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
