# Peer review of "The Use of Executive Fluency Tasks to Detect Cognitive Impairment in Individuals with Subjective Cognitive Decline"

_behavsci, 2022, doi:10.3390/bs12120491_

Round 1

Reviewer 1 Report

In this study 60 older adults with subjective cognitive decline and 60 healthy controls completed measures of verbal fluency with varying levels of cognitive demand. The SCD group was found to have worse performance only on the more difficult measures that required executive function. Responses were not correlated with education level or depression symptoms.

Overall, I think the manuscript addresses an important topic of interest, has a reasonable sample size, and presents the findings clearly. I only have a few minor points of clarification:

I think the objective/methods/hypothesis of the current study should be stated more clearly in the final paragraph of the introduction. It would be helpful to state which measures of verbal fluency (and not e.g. memory or standard executive function tasks) will be administered.

How long was the entire experimental session? Did all participants receive the tasks in the same order?

The orthographic constraint task is said to require inhibition, but it is not clearly explained how in the methods. I am also unsure if is this a new measure created by the authors, and what evidence they have that it does uniquely require inhibition. The lack of differences between groups in errors committed might indicate that it is not inhibition per se that is important, but simply the additional cognitive load.

Author Response

I think the objective/methods/hypothesis of the current study should be stated more clearly in the final paragraph of the introduction. It would be helpful to state which measures of verbal fluency (and not e.g., memory or standard executive function tasks) will be administered.

Response. First, we thank the reviewer for his/her very positive appreciation of our study. As requested, we added the information on the two experimental verbal fluency tasks in the final paragraph of the introduction

How long was the entire experimental session? Did all participants receive the tasks in the same order?

Response. This information was added on page 6

The orthographic constraint task is said to require inhibition, but it is not clearly explained how in the methods. I am also unsure if is this a new measure created by the authors, and what evidence they have that it does uniquely require inhibition. The lack of differences between groups in errors committed might indicate that it is not inhibition per se that is important, but simply the additional cognitive load.

Response. We thank the reviewer for this comment. We added information on the fact that this task is new and original. The additional demand this task has on inhibitory mechanisms was inferred from its nature, as any other cognitive task measuring executive functions. We modified the sentence related to the involvement of inhibition as follows: “Compared to typical fluency tasks, the orthographic constraint fluency task is assumed to require more inhibitory control abilities”(see p. 5).

According to us, the lack of differences between the two groups in terms of constraint errors cannot be interpreted as evidence for an absence of a deficit of inhibition. The SCD participants may still be able to inhibit responses but, this is more challenging for them and lead to the production of fewer responses as compared to the controls.

Reviewer 2 Report

This an important, well constructed and well-written study. People with subjective cognitive decline are an interesting group; they are at an increased risk of neurodengerative diseases, but it is difficult to identify any impairment in them objectively. The authors found differences between healthy controls and SCI when the fluency tasks require executive function (inhibition or flexibility).

The limitation section was very comprehensive and answered my remaining queries.

 My only suggestions to improve the paper would be to consider presenting the data in plots to provide greater clarity to the reader. Also you may consider citing a recent study (see Costa et al. https://doi.org/10.3390/geriatrics7040072) that demonstrated figure copying tests are not sensitive to changes between HC and SCI.   In regards to the areas of improvement, I suppose the figure could be a box and whisker plot - showing the mean + distribution of the fluency scores + errors scores for the HC and SCI groups. The fact that the difference in verbal fluency is apparent only during the fluency tasks that load on executive functions is a promising result.

Author Response

This an important, well constructed and well-written study. People with subjective cognitive decline are an interesting group; they are at an increased risk of neurodegenerative diseases, but it is difficult to identify any impairment in them objectively. The authors found differences between healthy controls and SCI when the fluency tasks require executive function (inhibition or flexibility).

Response. We thank the reviewer for his/her very positive appreciation of our study.

The limitation section was very comprehensive and answered my remaining queries.

 My only suggestions to improve the paper would be to consider presenting the data in plots to provide greater clarity to the reader. Also you may consider citing a recent study (see Costa et al. https://doi.org/10.3390/geriatrics7040072) that demonstrated figure copying tests are not sensitive to changes between HC and SCI.  In regard to the areas of improvement, I suppose the figure could be a box and whisker plot - showing the mean + distribution of the fluency scores + errors scores for the HC and SCI groups. The fact that the difference in verbal fluency is apparent only during the fluency tasks that load on executive functions is a promising result.

Response. We decided not to present the results in graphs to provide the reader with the entire data on the total score and number of errors, but also on the performance of the participants according to the time interval in each task. If the reviewer considers essential the addition of a figure illustrating the results, we will do it with pleasure.

We thank the reviewer for the suggestion regarding references and wee added the citation of Costa et al. as well as the citation of Tahmasebi et al. (see p. 3)